# CSR and Organizational Attractiveness: The Impacts of Crisis and Crisis Response

Wen-Ching Chang [1,*] , Liang-Chieh Weng [2] and Song-Bang Wu [1]

1   Department of Business Administration, Providence University, Taichung 43301, Taiwan
2   Department of International Business, Providence University, Taichung 43301, Taiwan
*   Correspondence: wcchang2@pu.edu.tw

**Abstract:** This study explores whether different sources of CSR information (i.e., the organization itself vs. the third party) and CSR reputation (i.e., leading vs. backward) affect job applicants' attraction to organizations. This study demonstrates the interaction effects of sources of CSR information and CSR reputation on organizational attractiveness and contributes to the literature by identifying the impact of crisis and crisis management strategies of the organization on its organizational attractiveness. From a Situational Crisis Communication Theory (SCCT), we examined the impacts of the crisis on organizations and what the crisis response strategies (i.e., excusing, apology, and scapegoating) organizations applied influence their organizational attractiveness. A fictitious food company was created for the experimental study. In total, 345 undergraduate business students at a university in central Taiwan were randomly assigned to 13 groups in different experimental settings. ANOVA and paired-sample *t*-tests were used to test the hypothesis. We found that (1) significant impacts made by the interaction effects of CSR reputations and the sources of CSR information of organizational attractiveness; (2) crisis events decreased organizational attraction dramatically regardless of the interaction of the sources of CSR information and CSR reputations; and (3) crisis management strategies effectively reduced the damages of crises on organizational attractiveness.

**Keywords:** corporate social responsibility (CSR); crisis; crisis response strategy; organizational attractiveness

## 1. Introduction

The organizational attractiveness to potential applicants has increasingly attracted the attention of recruitment practitioners and researchers [1–3]. Several factors that affect applicant attraction have been identified in the literature [3–5]. However, there is a comprehensive argument that corporate social responsibility (CSR) plays an increasingly important role in applicants' perceptions of their prospective organization. Forbrun and Shanley [6] and Turban and Greening [7] found that by executing social responsibility, companies may attract a better quality and higher quantity of job applicants by developing more positive images and a good reputation; thus, CSR was considered as a source of competitive advantage [6,8]. The current and prospective employees are reported as the vital audience in corporate responsibility communication [9]. A few empirical studies focus on the relationship between CSR and application attractions. For example, Bauer and Aiman-Smith [10] and Turban and Greening [7] examined a set of observable corporate social activities and policies designed to address social issues, and Zhang and Gowan [11] investigated the independent relationship of different aspects of CSR with applicant attraction. It shows that CSR can enhance organizational reputations [12] and has played an essential role in public relations [4].

Organizational crises create high levels of uncertainty and threats [13] and have potentially greater effects [14] within an organization. At the same time, crises are viewed as threats to organizational reputation. Not only do they damage the reputation, but

such shocks affect how stakeholders interact with the organization [15,16]. What organizations can apply to repair reputations and prevent reputational damage is post-crisis communication [17]. Situational Crisis Communication Theory (SCCT) was proposed by Coombs [18], which provides a framework to explain how to maximize the reputational protection afforded by post-crisis communication.

Although Coombs [19] offered different crisis response strategies to fix specific crises, the results of empirical studies were still inconsistent [20–22]. The researchers attempted to enrich the literature on the influence of CSR on applicants' attractiveness and to test the effect of information sources of CSR on applicants' attractiveness. It is because the source through which individuals learn about CSR is one of the factors influencing individuals' perceived sincerity to a company's motives of CSR activities which also may determine individuals' attitudes toward a company's CSR activities and its image [12]. This study also explores the impacts of crisis and the effects of crisis response strategies on applicant attractiveness. This study will help organizational managers by providing evidence for the demand to maintain and employ proper CSR and crisis management.

## 2. Literature Review

### 2.1. Sources of CSR Information

As Friestad and Wright [23] noted, consumers learn more about companies' marketing strategies and tactics, including CSR, which affects consumers' choices. Still, companies have options regarding the causes they support and how they deliver this information. These choices may not necessarily reflect a genuine interest in the cause. This idea echoes the observation that brand reputations decline when consumers perceive manipulative intention from firm actions [24]. Thus, if other contextual information is available, consumers may process it systematically to determine the company's true motives. For example, Syzkman et al. [25] found that when consumers notice that the advertisement for "not drinking and driving" is sponsored by a beer company, they will feel that the company has more self-interested motives than the identical advertisement provided by a non-profit organization. Consumers expect to learn about the CSR activities of the organization itself and fair media sources, while some independent organizations provide relatively unbiased information on corporate CSR activities [26]. The source of information about CSR is seen as the factor in how the message is received [27–29]. Campbell and Kirmani [30] pointed out that individuals believe that companies have conducted CSR activities with insincere motives when they make proactive advertisements for their CSR activities. Suppose consumers perceive the company as carrying out CSR activities with ulterior motives; in that case, the company will receive counterproductive results to the executed CSR activities and may receive more negative reviews than those without any CSR activities. The corporate image or reputation will be negatively affected because consumers perceive the corporate intentions as manipulating consumer perception [30].

On the other hand, individuals infer that the company has a sincere interest in CSR activities when they receive this information from a fair media source or a third party [31], which aligns with Groza et al.'s study [32]. That is, proper communication of CSR can be a feasible way to instill positive corporate reputations and purchase intentions. However, Skard and Thorbjørnsen [33] revealed a contingency of source effects in communicating social sponsorships on the brand's pre-existing reputation. Based on the literature, we employed two information sources (the organization itself or a third party) and two kinds of CSR reputation (leading vs. backward) to create a $2 \times 2$ experimental design, by which we had four experimental settings presenting the interactions of CSR information sources and CSR reputation. The current study thus proposed the following hypothesis.

**Hypothesis 1 (H1).** *Participants of these four experimental settings (A, B, C, and D) will have different extents of perceptions of organizational attractiveness.*

The four settings are:

A: Leading CSR reputation* CSR information from the organization itself;
B: Leading CSR reputation* CSR information from the third party;
C: Backward CSR reputation* CSR information from the organization itself;
D: Backward CSR reputation* CSR information from the third party.

*2.2. Crisis*

A crisis is a sudden, unexpected, but not random event that evolves over time and is the product of a series of intentional or unintentional events that threaten to damage the operation of and to cause both a financial and reputational threat to an organization [31]. Crises would be the result of a cumulative and continuous organizational dysfunctions process [34], and they may cause physical, emotional, or financial harm to stakeholders in different aspects [35]. The public considers organizational reputation a valuable intangible asset that can attract customers, outstanding talents, investor interest, improve financial performance, and increases returns on investments, thereby creating a competitive advantage and receiving positive evaluations from financial analysts [31]. Tǎnase [25] noted that a crisis might strongly impact the organizational reputation or image. Following this idea, we proposed the following hypothesis.

**Hypothesis 2 (H2).** *The extent of participants' perceptions of organizational attractiveness will decline in the affection of a crisis in four experimental settings (A, B, C, and D).*

*2.3. Crisis Respond Strategy*

The crisis is an adverse event that causes stakeholders to assess crisis responsibility for the crisis and change their perception of organizations. Coombs [18] proposed the Situational Crisis Communication Theory (SCCT) based on the attribution theory. It predicts a crisis's reputational threat and advises crisis response strategies to protect reputational assets. Attributions of crisis responsibility from stakeholders shape their emotional and behavioral consequences for the organization [17]. Once the organization is accounted for responsibly, its reputation suffers, and stakeholders are displeased. Stakeholders may lose their connection to the organization, which leads to lousy word-of-mouth about the organization. Managers are responsible for preventing either of these two adverse outcomes [35], which means the strategies/actions taken in response to a crisis are critical to maintaining and protecting the organization's reputation. Crisis response strategies are defined as the actions and statements that organizations make after crises, and whether it is helpful or harmful to the recovery of the organizational reputation relies on the quality of the crisis response [21].

Coombs [19] proposed ten crisis response strategies: attacking the accuser, simple denial, scapegoating, excusing, justification, compensation, apology, reminding, ingratiation, and victimage. Choosing an appropriate crisis response strategy would affect the public's attribution of responsibility for the crisis [35]. SCCT has been applied in several studies to test whether specific crisis response strategies work for certain crisis types [20,22]. For example, Sisco et al. [22] noticed that crisis impacts could be resolved more effectively through certain crisis response strategies. However, Ki and Brown's [21] study yielded a different result, showing that none of the tested crisis response strategies helped to reduce public blame for a particular responsibility of an organization in the crisis. There have been several severe crises over the last several years in Taiwan's food industry. The response strategies employed by these companies, including scapegoating, apology, and excusing, had varying levels of effectiveness. Because of the inconsistent findings of previous studies about crisis response strategies, the following hypotheses will be tested on the effects of

different selected response strategies, which are used in real cases after a crisis happened to a food company.

**Hypothesis 3.1 (H3.1).** *Participants' perceptions of organizational attractiveness will be higher after the effective implementation of crisis response strategies compared to perceptions in the immediate aftermath of a crisis.*

**Hypothesis 3.2 (H3.2).** *The extent of participants' perceptions of organizational attractiveness will be significantly different when companies employ different crisis response strategies (scapegoating, apology, and excusing).*

### 3. Methods

A fictitious food company was created for the experimental study instead of using a real company as research material to avoid any bias based on an existing company. People participate in the purchase and consumption of food on a regular basis. Therefore, a food company was designed as the fictitious company in the scenario due to a high degree of familiarity with the food industry across the research sample. The researchers felt that this familiarity might facilitate higher and more knowledgeable participation among the population of the sample. The study investigates factors influencing organizational attractiveness, focusing on the interaction of CSR information sources and CSR reputation, crises, and crisis response strategies. Since the company's operation is dynamic, this research attempts to observe whether the organizational attractiveness will change dynamically due to different events and the coping methods adopted. The conceptual research framework is shown in Figure 1.

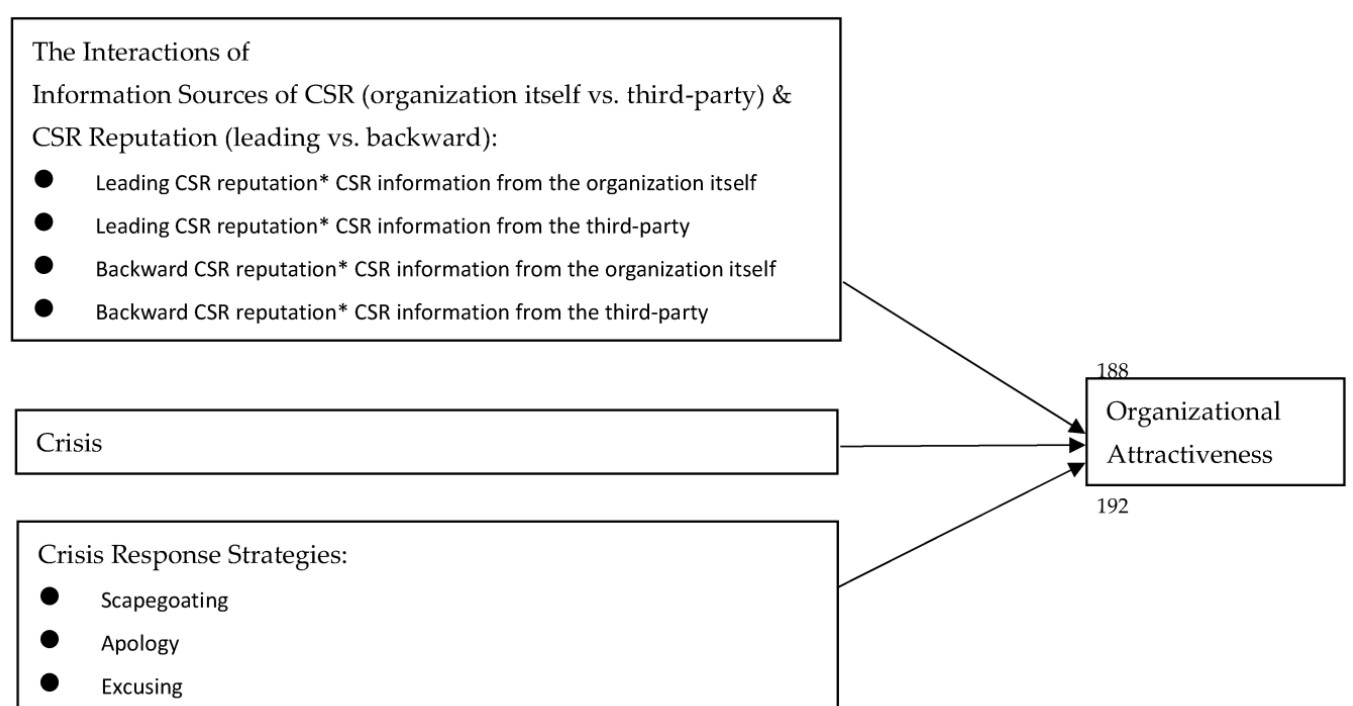

**Figure 1.** Research conceptual framework.

The subjects of this study were university graduates-to-be in central Taiwan. This study used the Commonwealth Magazine Corporate Citizenship Award, which is familiar to people in Taiwan, as the reference for CSR evaluation. In this study, CSR performance evaluation of the experimental settings are set as leading (the third among 500 enterprises) and backward (the 470th among 500 enterprises). The sources of information are divided into two types: the organization itself (from the company website) and the neutral third-party source (from the Commonwealth Magazine). A 2 × 2 experimental design was adopted, as shown in Table 1, which contained four settings of different scenarios in step 2. Three crisis response strategies were employed in the experimental setting: scapegoating, apology, and excusing.

**Table 1.** The flow of experiment and experimental setting arrangements.

| | Participants | | Step 1 | | Step 2 | | Step 3 | | Step 4 | |
| | M | F | INTRO | OAS | E and IS | OAS | Crisis | OAS | CRS | OAS |
|---|---|---|---|---|---|---|---|---|---|---|
| G1 | 10 | 16 | ✔ | ✔ | | | | | | |
| G2 | 7 | 20 | ✔ | ✔ | A | ✔ | ✔ | ✔ | X | ✔ |
| G3 | 6 | 20 | ✔ | ✔ | A | ✔ | ✔ | ✔ | Y | ✔ |
| G4 | 9 | 18 | ✔ | ✔ | A | ✔ | ✔ | ✔ | Z | ✔ |
| G5 | 6 | 21 | ✔ | ✔ | B | ✔ | ✔ | ✔ | X | ✔ |
| G6 | 9 | 18 | ✔ | ✔ | B | ✔ | ✔ | ✔ | Y | ✔ |
| G7 | 4 | 23 | ✔ | ✔ | B | ✔ | ✔ | ✔ | Z | ✔ |
| G8 | 6 | 20 | ✔ | ✔ | C | ✔ | ✔ | ✔ | X | ✔ |
| G9 | 12 | 14 | ✔ | ✔ | C | ✔ | ✔ | ✔ | Y | ✔ |
| G10 | 3 | 23 | ✔ | ✔ | C | ✔ | ✔ | ✔ | Z | ✔ |
| G11 | 8 | 18 | ✔ | ✔ | D | ✔ | ✔ | ✔ | X | ✔ |
| G12 | 7 | 20 | ✔ | ✔ | D | ✔ | ✔ | ✔ | Y | ✔ |
| G13 | 7 | 20 | ✔ | ✔ | D | ✔ | ✔ | ✔ | Z | ✔ |

Note: OAS = organizational attraction scale; E and IS = CSR reputation and CSR sources of information; CRS = crisis response strategy. A: Leading CSR reputation* CSR information from the organization itself; B: leading CSR reputation* CSR information from the third party; C: backward CSR reputation* CSR information from the organization itself; D: backward CSR reputation* CSR information from the third-party; X: scapegoating; Y: apology; Z: excusing.

### 3.1. Participants

The third author of this study distributed and collected written survey materials in the respondents' classrooms during regular university hours. Participants were fully informed about the purpose of this study. The researchers gave each participant a package containing instructions, surveys, and scenario cases. Respondents had to read the scenarios and imagine themselves as job seekers willing to apply for a job at the case company. All participants were divided into 13 groups reading different scenarios randomly and equally (see Table 1). Each group included about 26 or 27 participants. A total of 345 senior undergraduate business students at a university in central Taiwan participated in this research. The distribution of the age groups was 78.8% under 22 years old, 17.1% 23 years old, 2.9% 24 years old, and 1.2% older than 25 years old. Men comprised 27.2% of the sample.

### 3.2. Procedure

The third author on this paper gave a brief introduction of this research project and distributed the scenarios to all participants in several undergraduate classes in management school. There are 4 steps in this experiment. In the first step all participants read the same scenario about the introduction to the company. The effect of CSR on organizational

attractiveness was tested on this step. All participants (except Group 1) were assigned to 4 settings (A, B, C, and D) of the combinations of CSR reputation (leading vs. backward) and the information sources of CSR (from the company itself vs. from the third party) in step 2. Hypothesis 1 was investigated in this step. Participants received the same crisis scenario in step 3 and the impact of the crisis event on job seekers' perception of organizational attractiveness was tested. Participants were assigned 3 scenarios of different response strategies. The effect of crisis response strategies on job seekers' perceptions of organizational attractiveness was examined, and the differences of the effects of these three crisis response strategies on participants' perceptions was studied in step 4. The detail of each step is explained as follows:

Step 1: All participants in every group read the same scenario involving the fictional company. The participants then immediately answered the organizational attractiveness scale.

Step 2: The interaction effects of CSR reputation (leading/backward) and the CSR information sources (from the organization itself/from a third party) on participants' perceptions of organizational attractiveness were tested in this step. The scenario for CSR reputation was using the most popular Corporate Citizenship Award in Taiwan hosted by Commonwealth Magazine which is the most well-known professional business magazine. We used two descriptions to describe one leading CSR company (an organization that placed third among 500 companies for CSR) and one company not known for effective CSR (placing 470th out of 500 companies). Two descriptions are used to separate CSR information sources placed by the organization itself (information posted on company website) and information from the third party (reported by Commonwealth Magazine). All participants answered the organizational attractiveness scale after reading these descriptions. The followings are the four scenarios:

A: Leading CSR reputation* CSR information from the organization itself;
B: Leading CSR reputation* CSR information from the third-party;
C: Backward CSR reputation* CSR information from the organization itself;
D: Backward CSR reputation* CSR information from the third-party.

Step 3: All participants except Group 1 read a scenario describing a serious food safety crisis in the company. Participants of Groups 2–13 answered the organizational attractiveness scale after reading the crisis description.

Step 4: Groups 2–13 were assigned to read three different crisis response strategies scenario descriptions ("scapegoating" assigned to setting X; "apology" assigned to setting Y, and; "excusing" assigned to setting Z), and then answered the same organizational attractiveness scale.

## 4. Results

This research employed one-way ANOVA to test the first hypothesis. Table 2 showed that the effect of the interaction of CSR reputation (leading/backward) and the information source of CSR (from organization itself/from third party) is significant ($F = 23.79$, $p < 0.05$). Thus, Hypothesis 1 was supported. The post hoc tests revealed that setting A (m = 3.56) has a significantly higher score than settings C (m = 3.02) and D (m = 3.08); further, setting B (m = 3.61) has a significantly higher score than settings C (m = 3.02) and D (m = 3.08) (Table 3). The results revealed the effects of perceived leading CSR practices on organizational attractiveness.

**Table 2.** Results of one-way ANOVA for testing the interaction effects of CSR reputation and CSR sources of information on organizational attractiveness.

| Sources | SS | df | MS | *F* |
|---|---|---|---|---|
| Between group | 23.473 | 3 | 7.824 | 23.790 * |
| Within group | 103.601 | 315 | 0.329 | |
| Sum | 127.073 | 318 | | |

* $p < 0.05$.

**Table 3.** Post hoc comparisons of CSR reputation and CSR sources of information on organizational attractiveness.

| I | J | Mean | Mean Difference (I-J) | SE | p Value | 95% Confidence Interval | |
|---|---|---|---|---|---|---|---|
| | | | | | | Lower Bound | Upper Bound |
| A | B | 3.612 | −0.052 | 0.090 | 0.953 | −0.306 | 0.202 |
| | C | 3.015 | 0.545 * | 0.091 | 0.000 | 0.288 | 0.801 |
| | D | 3.078 | 0.482 * | 0.091 | 0.000 | 0.228 | 0.737 |
| B | A | 3.560 | 0.052 | 0.090 | 0.953 | −0.202 | 0.306 |
| | C | 3.015 | 0.597 * | 0.091 | 0.000 | 0.341 | 0.853 |
| | D | 3.078 | 0.534 * | 0.090 | 0.000 | 0.281 | 0.789 |
| C | A | 3.560 | −0.545 * | 0.091 | 0.000 | −0.801 | −0.288 |
| | B | 3.612 | −0.597 * | 0.091 | 0.000 | −0.853 | −0.341 |
| | D | 3.078 | −0.063 | 0.091 | 0.927 | −0.319 | 0.194 |
| D | A | 3.560 | −0.482 * | 0.091 | 0.000 | −0.737 | −0.228 |
| | B | 3.612 | −0.534 * | 0.090 | 0.000 | −0.789 | −0.281 |
| | C | 3.015 | 0.063 | 0.091 | 0.927 | −0.194 | 0.319 |

* $p < 0.05$. A: Leading CSR reputation* CSR information from the organization itself; B: leading CSR reputation* CSR information from the third party; C: backward CSR reputation* CSR information from the organization itself; D: backward CSR reputation* CSR information from the third party.

Paired-sample *t*-test was used to analyze if the extent of organizational attractiveness declined significantly after crisis (shown in Table 4). The result indicated significant differences of organizational attractiveness before and after crisis ($t = 19.08$, $p < 0.001$), and the mean of the extent of organizational attractiveness before crisis ($\bar{x} = 3.32$) was significantly higher than after crisis ($\bar{x} = 2.35$). In sum, the extent of participants' perceptions of organizational attractiveness declined after crisis happened, no matter what CSR reputation (leading/backward) and the information source of CSR (from the organization itself/from a third party) the company has. This evidence supported Hypothesis 2.

**Table 4.** Paired-sample *t*-test for the impacts of crisis on organizational attractiveness.

| Variable | Team | No. of Samples | Mean | S.D. | t |
|---|---|---|---|---|---|
| Organizational attractiveness | Before crisis | 319 | 3.32 | 0.63 | 19.08 *** |
| | After crisis happened | 319 | 2.35 | 0.80 | |

*** $p < 0.001$.

Paired-sample *t*-test was used to analyze if the extent of organizational attractiveness increased significantly after the organization employed crisis response strategies (shown in Table 5). The result indicated significant differences of organizational attractiveness after crisis response strategies employed by the organization ($t = -8.14$, $p < 0.000$), and the mean of the extent of organizational attractiveness after crisis response strategies employed ($\bar{x} = 2.65$) was significantly higher than after the crisis happened ($\bar{x} = 2.35$). Hypothesis 3.1, thus, was supported. The means and standard deviations of each crisis response strategy are excusing (M = 2.589, sd = 0.67); apology (M = 2.73, sd = 0.65); and scapegoating (M = 2.62, sd = 0.68). Table 6 showed the result of one-way ANOVA to test the effect of different crisis response strategies. The result did not show support for Hypothesis 3.2 ($F = 1.216$, $p > 0.05$).

**Table 5.** Results of paired-sample *t*-test for testing the effects of crisis response strategies on organizational attractiveness.

| Variable | Team | No. of Samples | Mean | S.D. | *t* |
|---|---|---|---|---|---|
| Organizational attractiveness | After crisis happened | 319 | 2.35 | 0.80 | −8.14 *** |
| | After crisis response strategy | 319 | 2.65 | 0.67 | |

*** *p* < 0.001.

**Table 6.** Results of one-way ANOVA for testing the effects of crisis response strategies.

| Sources | SS | df | MS | *F* |
|---|---|---|---|---|
| Between Group | 1.08 | 2 | 0.542 | 1.216 |
| Within Group | 140.86 | 316 | 0.446 | |
| Sum | 141.95 | 318 | | |

## 5. Discussion

A growing number of studies suggest that an organization's CSR practices can affect its attractiveness as an employer, but the effects of the performance or evaluation of CSR is still rare. In particular, many enterprises participate in contests of social citizenship or CSR as evidence of their engagement of CSR, but whether the impact of being one of the winners awarded by these contests may capture the eyes of job seekers are still unknown. This research, therefore, investigates the effect of the evaluation result of CSR on organizational attractiveness. At the same time, information sources leading customers to have different ideas of the motivation of organizations engaging in CSR practices was proved by research. The current research employed this variable to test the combination effects of information sources and CSR evaluation on organizational attractiveness.

In the sequential experiments, we designed three manipulations, which are the interactions of reputations and sources of information about CSR, the crisis that happened in these companies, and the crisis response strategies used by these companies. The results of first manipulation provide evidence of the power of the interaction of evaluation and information sources about CSR on organizational attractiveness. The statistical results indicating the effects of setting A (Leading CSR reputation* CSR information from the organization itself) and setting B (Leading CSR reputation* CSR information from the third-party) are significantly higher than setting C (Backward CSR reputation* CSR information from the organization itself) and setting D (Backward CSR reputation* CSR information from the third party). Although, based on the research design, the solo effects of the source of CSR information could not be identified, the current study showed that an organization with a leading CSR reputation has relatively higher organizational attractiveness regardless of the information sources. Skard and Thorbjørnsen [28] indicated inconsistent findings on the effects of sources on advertising versus publicity among previous studies.

Previous studies have indicated that CSR information delivered through neutral third-party sources will be evaluated more favorably than corporate sources [36]. The current research results are not consistent with the previous research and are also different from the concept of Skard and Thorbjørnsen [28]; that is, the contingency of the source effect in the communication of social sponsorship has an impact on the brand's pre-existing reputation. More specifically, the publicity generated more positive brand evaluations in the high-reputation condition than advertising. In the low-reputation condition, advertising generated more positive brand evaluations than publicity [28]. In contrast, our findings are different from previous studies. Whether the CSR information source comes from the organization itself or an impartial third party, a leading CSR reputation can generate more positive organizational attractiveness than a backward CSR reputation. The possible reason is that the CSR reputation we used resulted from a well-known competition in a credible local management magazine. This competition has been held for 16 years and is recognized and reliable in the local area. The impression formed by it is too strong to have

a decisive effect on the participants' perception of organizational attractiveness. The results of this study inspire us that the affirmation of an organization's investment in CSR by a well-known and credible institution significantly impacts it. Through the endorsement and affirmation of these institutions, no matter what channel the CSR reputation information is delivered to the applicants, it can affect their perceptions of organizational attractiveness.

In the second operation, we observed participants' responses to the crisis in the case company, which showed that the crisis significantly negatively impacted participants' perception of organizational attractiveness regardless of experimental settings A, B, C, and D. Crises have the potential to damage a company's reputation and reduce job seekers' attractiveness to the organization. These findings serve as reminder that companies should be more conscientious in preventing crisis events. A crisis event may cause a company to lose competitive advantage, market share, and other more extreme consequences. Prevention is always better than treatment. A crisis maybe happen suddenly, but never accidently. It would be the result of a cumulative organizational dysfunctions process over time and the product of a series of intentional or unintentional events [17,34]. Therefore, perfect process planning and actual implementation to maintain the smoothness and correctness of daily operations are the key elements to ensure that the crisis does not occur.

The last manipulation is about the impacts of the crisis response strategies employed by the case company on organizational attractiveness; we found that all types of response strategies successfully pooled up participants' perceptions of organizational attractiveness. However, an effective crisis response strategy can decrease the impacts on a company once a crisis is underway. The current study found that no matter what crisis response strategy (scapegoating, apology, and excusing) is taken, it is always better than doing nothing. The crisis literature regards crisis response content as a rhetorical strategy, and the consensus in the literature is that the effectiveness of rhetoric will decrease with the increase in organizational responsibility attribution [37]. When the public's attribution of responsibility to the organization is higher, it must adopt proper strategies to restore its image [38]. On the other hand, in the "transgressions" type of crisis, if the organization can show compassion and empathy, then the crisis communication will positively impact its image [39]. Accordingly, specific response strategies should be appropriate for specific crisis types. However, Huang's [37] research pointed out that the crisis response strategy includes the content and the communication form. The research found that positive response forms, which refer to how an organization conducts or executes a crisis communication strategy, were more likely to create positive organizational and public relations than crisis communication strategies. These findings explain why there are no significant differences in the effects of the three crisis response strategies, while the current study investigated the crisis response strategy focusing on communication content.

## 6. Limitations and Further Research

The current study clarified the relationships among CSR reputation, crisis management, and organizational attractiveness which enriches the literature of CSR and HRM relevant research; however, the sample of this study were undergraduate and graduate students, which indicates the limited generalization of these research results. Because of the homogeneity of samples, this research study was unable to investigate the effects caused by demographic variables. Therefore, we suggest that further research may employ the general public as research samples to obtain their cognition of intentions and evaluation of companies' CSR activities, and the impacts of demographic variables. Secondly, the current study investigates the interaction effect of CSR reputation and source of information and, thus, we are unable to identify the solo effect of source of information. In the future, we recommend identifying the simple effect of source of information which can provide clear picture of the importance of information source. Thirdly, the powers of different crisis strategies were not identified in the current study; however, it is worth it to figure out the matches of crisis types and response strategies. We recommend future research can take

one step further to investigate the effect of crisis strategy, including communication content and forms, for specific type of crisis.

## 7. Conclusions

This study differs from previous studies in two aspects. First, although there are many related studies on the factors that affect the attractiveness of an organization [40–44], there are few studies on the impact on the attractiveness of an organization when it encounters a crisis. Furthermore, in terms of empirical research, there are few opportunities to study the perception of potential applicants before and after the crisis. This study adopted a quasi-experimental design through the scenario to explore the changes in potential job applicants' perception of organizational attractiveness before and after the crisis event in the organization, which is also one of the characteristics of this study. Previous studies aimed at Coomb's SCCT theory [17,19,35] that specific crisis response strategies are suitable for certain crises had inconsistent results [20–22]; this study added an unsupported result to the literature. In addition, most of the previous relevant research focused on consumers or customers. It explored the impact of organizational crisis response strategies on consumers, customers, or the general public's perception of the organization's image. The research participants of this study were potential job applicants, and the research results can be used as the basis for human resource management personnel in managing employer brands. The critical finding from this study is the significant impact that CSR reputation can have on prospective employees seeking employment, especially if the CSR reputation is confirmed by a credible institution, which validates the value of the investment in CSR practices. The verification power of CSR reputation confirmed by trusted institutions dominates the interaction between CSR information sources and CSR reputation. This finding brings a different perspective to CSR communication and marketing research.

Several notable implications emerged based on the findings. First, our results extend the role of reputation, information source, crisis, and crisis response strategies in explaining job applicants' responses to CSR activities. While previous research has studied the effects of CSR-relevant activities on various outcomes, the current research investigated the CSR reputation and sources of information, suggesting that business executives' concern about how to derive substantial benefits from CSR investments. As long as talents, then, are important elements of competitiveness, an organization can attract more quality talents through their positive reputation by investing in CSR. Research indicated the positive impacts of CSR on organizational reputation [10,42,44–46]. The image of responsible corporate citizen earned by a company via CSR can bring others benefits, such as increasing support from stakeholders, improving brand image, increasing reputation, increasing purchasing intentions from customers, and attracting more highly qualified talents [10,42,44,45,47]. In summary, the affirmation received by enterprises for their CSR activities has a significant positive effect on organizational attractiveness. However, business operations will encounter various crises to some extent, and the damage to organizational attractiveness caused by the emergence of crises is inevitable. The crisis response strategies adopted after the crisis can effectively alleviate the impact of organizational attractiveness.

**Author Contributions:** W.-C.C., conceptualization, writing—original draft preparation, writing—review and editing, supervision, funding acquisition; L.-C.W., methodology, software, validation, resource, supervision; S.-B.W., formal analysis, investigation, data curations, visualization, project administration. All authors have read and agreed to the published version of the manuscript.

**Funding:** This research is a part of an integrated research project funded by the National Science Council of Taiwan (NSC 101-2632-H-126-001-MY3).

**Institutional Review Board Statement:** Not applicable.

**Informed Consent Statement:** Not applicable.

**Data Availability Statement:** Not applicable.

**Acknowledgments:** The authors are grateful to the editor and the anonymous reviewers of this paper.

**Conflicts of Interest:** The authors declare no conflict of interest.

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
