# Peer review of "CSR and Organizational Attractiveness: The Impacts of Crisis and Crisis Response"

_sustainability, doi:10.3390/su15043753_

Round 1

Reviewer 1 Report

This is a simple study of factors affecting organizational attractiveness. Although the study is simple, I think it is still worth publishing if you can address my comments.

1.       Lines 37-41: This sentence is not clear.

2.       Lines 53-61: This part is not related to organizational crisis. You should start a new paragraph from “Although Commbs …” (line 51).

3.       Line 82: What do you mean by 2*2 interaction combinations of CSR reputation (leading/backward)? You have not discussed this in the text. Moreover, please explain how the combinations affect participants’ perceptions.

4.       Line 136: In your H3.2, “diff” is a short form of “difference”. So, please write “different” instead of “diff”.

5.       Your hypotheses are very complicated. Is it possible for you to draw a model to illustrate the relationships among the variables? Alternatively, you should consider making them concisely. Unnecessary words or parts of the sentences should be removed.

6.       In Table 1, you should include a column of number of female participants for each of the thirteen groups and the second column for male participants.

7.       As you used G for excusing, you should not use G for group number. Alternatively, you may keep using G for group number, and change EFG to say XYZ.

8.       Lines 215-220: This is a long sentence, which is very complicated. Revise it to make the findings clear in presentation.

9.       Table 3: Although in general H1 is supported, it is clear that settings A and B (as well as C and D) are not significantly different. So, there is only the effect of reputation, not the source of information. In your discussion section, you have not explored why the source of information was not a significant factor.

10.   Lines 241-2: The mean after crisis response strategies should be 2.65.

11.   Lines 294-296: Please remove “no matter what combination … the company has”. This would mislead readers that reputation and information source have been considered when testing the effect of crisis on attractiveness.

12.   You have not discussed why crisis response strategies were not a significant factor.

13.   No mentioning of any limitations of the study.

Author Response

Thank you so much for your review and insightful comments that helped us reconsider our research and improve the quality of our manuscript.  We have learned a lot from this revision opportunity. 

We answered all your questions and modified our manuscript accordingly.  

Some paragraph marked in yellow is especially for answering your questions and requests. Please see our answers in the attached file. 

Reviewer 2 Report

This article investigates the effects of sources of CSR information and CSR reputation on organizational attractiveness. First comment: very short.

It exposes the impacts of the crisis in an organization and the effects of different crisis response strategies

The literature review addresses several interesting topics and hypothesis.

The table 1 does not contribute to the article and seems incomplete. You should delete it.

The authors collected data from a sample of 345 senior undergraduate business students at a university in central Taiwan.

The results are very interesting but there is too much tables. You should rethink about it.

The theoretical and managerial contributions could be improved. Moreover, no limitations are highlighted.

I had a great interest and pleasure in reading this first version of the article, may the authors be thanked and encouraged.

I give a favorable opinion.

Author Response

Thank you so much for your review and insightful comments.  

We appreciate this opportunity to revise our manuscript. 

We answer all your questions and update our manuscript accordingly.  

Please see our answers in the attached file.  

Thank you.

Round 2

Reviewer 1 Report

Thank you for addressing my concerns. The following are my final comments for you:

1. Line 214: It should be Table 1, not Table 2.

2. Line 209: I computed that the total number in Table 1 should be 344. But in line 209, you stated that the total number is 345. Which one is correct?

Author Response

Dear reviewer,

Thank you so much for your kind review and reminder. Your review helps us to improve our manuscript so much. We also sent this manuscript to do language editing.  Some minor changes marked in red were made accordingly.
